# Topical *Artocarpus communis* Nanoparticles Improved the Water Solubility and Skin Permeation of Raw *A. communis* Extract, Improving Its Photoprotective Effect

**DOI:** 10.3390/pharmaceutics13091372

**Published:** 2021-08-31

**Authors:** Chun-Yin Yang, Pao-Hsien Huang, Chih-Hua Tseng, Feng-Lin Yen

**Affiliations:** 1School of Pharmacy, College of Pharmacy, Kaohsiung Medical University, Kaohsiung City 807, Taiwan; anitayang0812@gmail.com (C.-Y.Y.); j6466497@gmail.com (P.-H.H.); 2Drug Development and Value Creation Research Center, Kaohsiung Medical University, Kaohsiung City 807, Taiwan; 3Department of Fragrance and Cosmetic Science, College of Pharmacy, Kaohsiung Medical University, Kaohsiung City 807, Taiwan; 4Department of Medical Research, Kaohsiung Medical University Hospital, Kaohsiung City 807, Taiwan; 5Department of Pharmacy, Kaohsiung Municipal Ta-Tung Hospital, Kaohsiung City 801, Taiwan; 6Institute of Biomedical Sciences, National Sun Yat-Sen University, Kaohsiung City 804, Taiwan

**Keywords:** *Artocarpus communis*, topical antioxidant nanoparticles, water solubility, skin penetration, photocytotoxicity, cellular uptake

## Abstract

Antioxidants from plant extracts are often used as additives in skincare products to prevent skin problems induced by environmental pollutants. *Artocarpus communis* methanol extract (ACM) has many biological effects, such as antioxidant, anti-inflammatory, wound healing, and photoprotective effects; however, the poor water solubility of raw ACM has limited its applications in medicine and cosmetics. Topical antioxidant nanoparticles are one of the drug-delivery systems for overcoming the poor water solubility of antioxidants for increasing their skin penetration. The present study demonstrated that ACM-loaded hydroxypropyl-β-cyclodextrin and polyvinylpyrrolidone K30 nanoparticles (AHP) were successfully prepared and could effectively increase the skin penetration of ACM through changing the physicochemical characteristics of raw ACM, including reducing the particle size, increasing the surface area, and inducing amorphous transformation. Our results also revealed that AHP had significantly better antioxidant activity than raw ACM for preventing photocytotoxicity because the AHP formulation increased the cellular uptake of the ACM in UVB-irradiated HaCaT keratinocytes. In conclusion, our results suggest that AHP may be used as a good topical antioxidant nanoparticle for delivering ACM into deep layers of the skin for preventing UVB-induced skin problems.

## 1. Introduction

*Artocarpus communis* (synonym: *Artocarpus altilis*), which belongs to the family Moraceae, is a multi-purpose crop grown in southeast Asia and Taiwan. Its economic importance stems from its use in agriculture, folk medicine, and natural products. The ethnopharmacological uses of *A. communis* include the treatment of malarial fever, diarrhea, and infection. Previous studies have also mentioned that *A. communis* methanol extract (ACM) contains many phenolic compounds, including flavonoids, stilbenoids, arylbenzofurons, and jacalin [1], presenting many pharmacological activities, such as anti-inflammatory, antioxidant [2], wound healing [3], melanogenesis-inhibitory [4], and anti-cancer effects [5]. Raw ACM can easily dissolve in organic solvents, such as alcohol and DMSO, but it is practically insoluble in water due to the poor water solubility of the index ingredient artocarpin that is abundant in ACM. Unfortunately, organic solvents may cause several skin problems, including skin irritation, allergy, redness, and peeling. These disadvantages limit their applications in medicine and the cosmetic industry. Therefore, it is very important to use non-irritating compounds such as cyclodextrins and polymers to prepare ACM for external use and effectively deliver ACM into the skin layer, while preserving its biological activity.

Drug-delivery systems, such as nanoparticles [6,7], inclusion complexes [8], microemulsions [9], solid dispersion [10], and cocrystallization [11], are commonly used to resolve the poor water solubility of active ingredients for increasing their absorption and bioavailability. Nontoxic nanocarriers with good manufacturing processes are the first choice for producing safe nanoparticle formulations. 2-hydroxypropyl-beta-cyclodextrin (HPBCD), a cyclic oligosaccharide with a donut-like shape that consists of seven glucopyranose units, not only serves as a lipophilic component, interacting with the active ingredients for overcoming the water solubility issue [12,13], but also enhances their stability and masks the undesirable smell or taste [14]. In addition, polyvinylpyrrolidone K30 (PVPK30), a nontoxic water-soluble polymer with an average molecular weight of 30,000, is often used as a stabilizer or dispersant in nanoparticle formulations [15]. Our preliminary data indicate that HPBCD cannot be successfully used to prepare ACM nanoparticles, but HPBCD combined with PVPK30 allowed the easy production of ACM nanoparticles that easily dissolved in water, resulting in a clear and transparent solution (Figure 1). However, the mechanism by which the water solubility and skin penetration of ACM nanoparticles is improved and their biological activity have not yet been elucidated. 

In the present study, ACM-loaded PVPK30/HPBCD nanoparticles (AHP) were prepared using the solvent evaporation method. We compared the physicochemical characteristics of AHP and raw ACM using solubility tests, FTIR, SEM, TEM, and XRD for elucidating the mechanism of the water-solubility improvement. An ex vivo skin-penetration study was used to compare the skin absorption of AHP and raw ACM. For confirming the biological activities, the DPPH free-radical-scavenging ability, an in vitro photoprotective cell model, and cellular uptake were used to compare the antioxidant activity.

## 2. Materials and Methods 

### 2.1. Materials

Artocarpin was purchased from Pulin Biotech Company Limited (Taipei, Taiwan). 2-hydroxypropyl-β-cyclodextrin (HPBCD) was obtained from Zibo Qianhui (Shandong, China). Polyvinylpyrrolidone K30 (PVPK30; MW ≅ 40,000; K-value, 29–32) and 2,2-diphenyl-1-picrylhydrazyl (DPPH) were obtained from Sigma (St Louis, MO, USA). Dimethyl sulfoxide (DMSO), and methanol were purchased from Aencore Chemical (Surrey Hills, Australia). Potassium dihydrogen phosphate (KH_2_PO_4_) and dipotassium phosphate (K_2_HPO_4_) were purchased from Ferak (Berlin, Germany). 3-(4,5-cimethylthiazol-2-yl)-2,5-diphenyl tetrazolium bromide (MTT) was purchased from MDBio (Taipei, Taiwan). HaCaT keratinocytes were obtained from the Istituto Zooprofilattico Sperimentale della Lombardia e dell’Emilia Romagna (Brescia, Italy). Dulbecco’s modified Eagle’s medium (DMEM) was purchased from Himedia Laboratories (Mumbai, India). Fetal bovine serum (FBS) was purchased from Thermo Fisher Scientific (Waltham, MA, USA). A penicillin, streptomycin, and amphotericin B solution was purchased from Biological Industries (PSA; Connecticut, NE, USA). All the other chemical reagents were of analytical grade.

### 2.2. Preparation of Artocarpus Communis Methanol Extracts (ACM)

The heartwood of *A. communis* was kindly provided by the Tainan District Agricultural Research and Extension Station. The plant species was authenticated by Dr. Ming-Hong Yen of the Graduate Institute of Natural Products, College of Pharmacy, Kaohsiung Medical University, Kaohsiung, Taiwan. The heartwood of *A. communis* was chipped and dried in the shade at room temperature for 3 days. A total of 200 g of heartwood pieces of *A. communis* and 4 L of methanol were added to a flask and placed in an ultrasonic bath (Branson 5510, Emerson Electric, St. Louis MO, USA), and extraction was performed for 1 h. The process was repeated twice, and the products were filtered, concentrated, and freeze-dried; the *A. communis* methanol extract (ACM) was finally obtained.

### 2.3. HPLC Analysis of ACM

The calibration curves for artocarpin and raw ACM were obtained by HPLC analysis (LaChrom Elite L-2000, Hitachi, Tokyo, Japan). The HPLC system consisted of an L-2130 pump, L-2200 autosampler, and L-2420 (UV–vis) detector. Artocarpin and raw ACM were eluted in methanol using the Mightysil RP-18 GP column (250 × 4.6 mm, i.d., 5 μm, Kanto Corporation, Portland, OR, USA). The mobile phase consisted of methanol and water (9:1). The flow rate was kept at 1 mL/min, and the UV–vis detector wavelength was set at 282 nm. Artocarpin and raw ACM were dissolved in methanol and then diluted to a series of concentrations for HPLC analysis. The calibration curves for artocarpin and raw ACM showed good linearity (r^2^ > 0.999). For the quality control of ACM, artocarpin is the major index component of ACM, and its calibration curve, based on HPLC analysis, was used to confirm the artocarpin content of the ACM in each extraction batch. The standard calibration curve for ACM was used to calculate the ACM content in AHP and other experiments.

### 2.4. Preparation of ACM-Loaded PVPK30/HPBCD Nanoparticle (AHP)

In this study, AHP nanoparticles were prepared with various ratios of ACM:PVPK30:HPBCD (1:18:2, 1:18:10, and 1:18:20; *w*/*w*/*w*) by the solvent-evaporation method. Briefly, 50 mg of raw ACM was dissolved in 10 mL of ethanol; then, various ratios of HPBCD were added, and stirring was continued for 1 h. Next, different amounts of PVPK30 were dissolved in the ACM/HPBCD solution with continuous stirring for 1 h. Subsequently, the ethanol in the mixed solution was removed using a rotary vacuum evaporation system at 50 °C to obtain a dry, dark-orange, powder-like ACM-loaded PVPK30/HPBCD nanoparticle (AHP). For the following experiments, the AHP was stored in a moisture-proof container until use.

### 2.5. Water Solubility of Raw ACM and AHP

In brief, 1 mg of raw ACM and AHP (corresponding to 1 mg of ACM) was added to 1 mL of distilled water. Each sample was vigorously shaken with a mixer (Vortex-Gene 2, Scientific Industries, Bohemia, NY, USA) for 10 min and then filtered with a 0.45 μm syringe filter (13 mm Acrodisc^®^ syringe filters with GHP membrane; Pall Corporation, Port Washington, NY, USA)). All the samples were analyzed by the HPLC method as described above, and then the solubility of ACM was calculated. 

### 2.6. Yield and Encapsulation Efficiency

The yield and encapsulation efficiency of a drug-delivery system are the major indices for evaluating a successful manufacture process. For determining the yield of AHP, different-ratio AHP formulations were dissolved in methanol. Each sample was shaken using a mixer for 10 min. After that, the ACM concentration of each sample was analyzed by the above HPLC-analysis method for calculating the yield of AHP. For determining the encapsulation efficiency of AHP, different-ratio AHP formulations were reconstituted in distilled water and then shaken with a mixer for 10 min. After that, 200 μL of each sample was placed into a high-speed centrifugal filter device (Nanosep^®^ Centrifugal Devices with Omega™ Membrane molecular weight 10,000, Pall Corporation, Port Washington, NY, USA) and then centrifuged at 10,000 rpm (Centrifuge 5430 R; Eppendorf, Hamburg, Germany) for 10 min. The encapsulated part was retained on the upper side of the tube, while the non-encapsulated part of the AHP was collected on the lower side of the tube. The amount of ACM in each sample was determined by the above HPLC analysis method. The yield and encapsulation values were calculated using the following equations:$$\mathrm{Yield}\;\left(\%\right)=\frac{\mathrm{ACM}\;\mathrm{weight}\;(\mathrm{mg})}{\mathrm{theoretical}\;\mathrm{amount}\;\mathrm{of}\;\mathrm{ACM}\;(\mathrm{mg})}\times 100\%$$
$$\mathrm{Encapsulation}\;\mathrm{efficiency}\;\left(\%\right)=\frac{\mathrm{theoretical}\;\mathrm{amount}\;\mathrm{of}\;\mathrm{ACM}-\mathrm{free}\;\mathrm{amount}\;\mathrm{of}\;\mathrm{ACM}}{\mathrm{theoretical}\;\mathrm{amount}\;\mathrm{of}\;\mathrm{ACM}}\times 100\%$$

### 2.7. Morphology and Particle Size Analysis

The shape and surface morphology of raw ACM, AHP, a physical mixture, and excipients were observed by using a scanning electron microscope (SEM; Hitachi S4700, Hitachi, Tokyo, Japan). Each sample was coated with gold–palladium under an argon atmosphere using an ion sputter coater (Hitachi E-1045, Hitachi, Tokyo, Japan). For capturing the particle morphology of AHP, 1 mg of AHP was reconstituted in 1 mL of distilled water, diluted 100-fold, and immediately placed onto a 200-mesh copper grid. A 0.5% (*w*/*v*) concentration of phosphotungstic acid (Sigma, St Louis, MO, USA), for negative staining, was placed onto the sample. After that, the sample was photographed using a transmission electron microscope (JEM-2000EXII instrument; JEOL Co., Tokyo, Japan).

### 2.8. Determination of Crystallinity

The crystalline states of ACM and AHP were analyzed by powder X-ray diffractometry (XRD; Siemens, Munich, Germany) with Cukα radiation at 40 kV and a current of 40 mA. XRD was operated in a range of 2θ angles from 5° to 50° in continuous scan mode, and the scanning rate was set at 1°/min.

### 2.9. Determination of Intermolecular Interaction

Fourier transform infrared (FTIR) spectroscopy (ALPHA II FTIR spectrometer, Bruker, Billerica, MA, USA) was used to determine the functional groups and bonding formation of the raw ACM and nanoparticle system. For each sample preparation, raw ACM, AHP (1:18:10), the physical mixture (1:18:10), HPBCD, and PVPK30 were mixed with potassium bromide (Sigma, St Louis, MO, USA). Then, each sample was milled using an agate mortar and compressed into a thin tablet for FTIR analysis. The scanning range of FTIR was set at 400 to 4000 cm^−1^ for determining the functional groups.

### 2.10. DPPH Free-Radical-Scavenging Activity

The DPPH (1,1-diphenyl-2-picrylhydrazyl) radical-scavenging activity was tested according to Lin et al. [16]. For sample preparation, raw ACM and AHP were reconstituted with a series of concentrations (50–500 μg/mL) in distilled water. Then, 100 μL of each sample was mixed with 100 μL of 200 μM DPPH free radical in 96-well plates. After that, the mixture solution was incubated in darkness at room temperature for 30 minutes. The optical density (OD_517_) of each DPPH reaction solution at 517 nm was measured using a microplate spectrophotometer. The DPPH radical-scavenging activity was calculated as follows:$$\mathrm{DPPH}\;\mathrm{scavenging}\;\mathrm{activity}\;\left(\%\right)=\frac{{\mathrm{OD}}_{517}\;\mathrm{of}\;\mathrm{control}\;-{\mathrm{OD}}_{517}\;\mathrm{of}\;\mathrm{samples}}{{\mathrm{OD}}_{517}\;\mathrm{of}\;\mathrm{control}}\times 100\%$$

### 2.11. In Vitro Skin Penetration 

According to the European Cosmetic Toiletry and Perfumery Association (COLIPA) guideline standard protocol, the experimental pig skin was purchased from a local butcher and cut into an appropriate size (2 cm × 2 cm). The buffer solution contained 0.14 M NaCl, 2 mM K_2_HPO_4_, and 0.4 mM KH_2_PO_4_ (pH 7.4). To begin with, the receptor chamber of a Franz diffusion cell was filled with 3.7 mL of buffer solution, and then the pig skin and the donor chamber were placed in sequence. After that, 200 μL of 1 mg/mL of raw ACM and AHP (ACM:PVPK30:HPBCD at a 1:18:10 ratio containing 1 mg of ACM) reconstituted in distilled water was loaded onto the stratum corneum of pig skin. The time points for topical administration were set at 0.5, 1, 2, and 4 h. The Franz diffusion cell system was maintained at 32 °C under a circulating water bath and stirred with a stir bar at 600 rpm. After that, each stratum corneum of skin sample was obtained by 15 rounds of tape stripping using 3 M Transpore tape. The residual skin sample was placed on a heat plate at 85 °C, and the epidermal and dermal layers were separated with a scalpel and cut into small pieces. Then, each sample was placed in a tube containing methanol, and the ACM was extracted under a sonicator bath for 1 h. All the samples were filtered through a 0.45 μm filter and analyzed using the above HPLC method to determine the ACM content.

### 2.12. Cell Viability 

Briefly, HaCaT cells were cultured in DMEM (Himedia) containing 10% fetal bovine serum with 1% penicillin/streptomycin/amphotericin B antibiotic solution and incubated in 5% CO_2_ at 37 °C (Direct Heat CO_2_ Incubator, Thermo Fisher Science, Waltham, MA, USA). For determining the cell viability, 1.5 × 10^4^ HaCaT cells were seeded in each well in 96-well plates, allowing the cells to adhere for 24 h. Then, the HaCaT cells were treated with a series of concentrations of raw ACM in PBS or DMSO, and AHP in PBS. After 24 h of incubation, the culture medium was removed and the cells were washed with PBS twice. A 150 μL volume of 0.5% methyl thiazolyl tetrazolium solution was added into each well. After 3 h of incubation, the plate was placed on a microplate spectrophotometer (BioTek μQuant, Winooski, VT, USA) and the optical density (OD_550_) of each well at 550 nm was determined; the cell viability was calculated using the formula below:$$\mathrm{Cell}\;\mathrm{viability}\;\left(\%\right)=\frac{{\mathrm{OD}}_{550}\;\mathrm{of}\;\mathrm{samples}}{{\mathrm{OD}}_{550}\;\mathrm{of}\;\mathrm{control}}\times 100\%$$

### 2.13. Cellular Uptake

For comparing the cellular uptake of raw ACM and AHP, 4 × 10^5^ HaCaT cells were seeded into each well of a 12-well cell culture plate and incubated for 24 h. Each well was treated with 5 μg/mL of raw ACM and AHP (1:18:10) in serum-free DMEM for 3, 6, 9, and 24 h. After that, the culture medium was removed and the HaCaT cells were collected using 0.5% SDS; the ACM contents of the cell lysates (intracellular compartment) were extracted with methanol under a bath sonicator for 10 min. Each sample was filtered using a 0.45 μm syringe filter, and the ACM content was determined by the HPLC method, as described above. 

### 2.14. Photoprotective Activity of Raw ACM and AHP in HaCaT Cells

HaCaT cells were seeded in a black 96-well plate at a density of 1.5 × 10^4^ cells/well and incubated overnight. The HaCaT cells were pretreated with different concentrations of raw ACM and AHP (1:18:10) in serum-free DMEM for 3 h. Then, the culture medium was removed and the cells were washed with PBS twice; 100 μL of PBS was then immediately added, and the cells were irradiated with a single dose of UVB (230 mJ/cm^2^). After that, the PBS was removed and the HaCaT cells were treated with raw ACM and AHP (1:18:10) with serum-free DMEM for 24 h. After incubation, the cell viability was determined using the MTT assay, as conducted previously.

### 2.15. Intracellular Reactive Oxygen Species (ROS) Assay

For measuring the intracellular ROS production with raw ACM and AHP, 1 × 10^4^ HaCaT cells were seeded in a black 96-well plate and incubated overnight. Subsequently, the HaCaT cells were pretreated with raw ACM and AHP in water for 24 h. Then, 10 μg/mL dichlorodihydrofluorescein diacetate (DCFH-DA; Sigma, Tokyo, Japan) solution was added into each well, and the cells were incubated for 30 min before UVB irradiation (230 mJ/cm^2^) and then incubated for 1 h. The relative fluorescence intensity was determined using a fluorescent plate reader (BioTek, Winooski, VT, USA) at excitation and emission wavelengths of 485 and 528 nm.

### 2.16. Statistical Analysis

The experiments were performed in triplicate, and all the data are represented as mean ± standard deviation (SD) using Microsoft Excel 2010 (Microsoft Office, Microsoft Corporation, Redmond, WA, USA). The statistical significance of differences between the experimental groups was tested by one-way ANOVA with Tukey’s test using SPSS 19 (SPSS Inc., Chicago, IL, USA), where *p* < 0.05 was considered the significance level. 

## 3. Results

### 3.1. The Yield, Artocarpin Content, and Calibration Curve for ACM

The freeze-dried ACM sample appeared as a dark-orange powder, and its extraction yield was about 3.4%. Artocarpin was the index component in ACM and produced a characteristic peak at 8.4 min in the HPLC chromatogram (Figure 1A). In addition, the HPLC chromatogram of ACM also had the same characteristic peak of artocarpin at 8.4 min and a significantly higher peak area than the other components in ACM. Our results indicate that the content of artocarpin in ACM was 67%. Moreover, the artocarpin calibration curve, showing a good linear fit (r^2^ > 0.999), was used to quantify the artocarpin content of the ACM, which was found to be 197 µg/mg. The present study also used methanol as a solvent to dissolve ACM with a series of concentrations to produce a calibration curve for ACM with good linearity (r^2^ > 0.999) for calculating the concentration of ACM in each formulation and for other experiments. 

### 3.2. Water Solubility, Yield, and Encapsulation Efficiency of AHP

As shown in Table 1, the water solubility of raw ACM was 3.5 µg/mL. USP pharmacopeia classifies a water solubility of a drug lower than 100 µg/mL as practically nil; thus, ACM was a practically insoluble component. By contrast, ACM was easily dissolved in organic solvents, such as ethanol, methanol, and DMSO. Table 1 also shows that the ACM:PVPK30:HPBCD mixtures at 1:18:2, 1:18:10, and 1:18:20 effectively increased the water solubility of raw ACM by 138.4, 289.2, and 188.3 times, respectively. The results imply that the water-solubility improvement of ACM depends on the ACM/PVPK30/HPBCD ratio; however, the highest ratio of 1:18:20 could limit the reconstitution of ACM in water. In addition, Table 1 shows that AHP (1:18:10) had the best yield and encapsulation efficiency among the three formulations. These results indicate that HPBCD with PVPK30 co-carriers can be used to effectively encapsulate ACM into a nanoparticle system, and AHP at a ratio of 1:18:10 might be an optimal formulation with which to overcome the poor water solubility of raw ACM and for use in other experiments.

### 3.3. Morphology of ACM and AHP

Figure 2 clearly illustrates the SEM microphotograph of the ACM, the excipients, the physical mixture, and AHP. It can be seen that the raw ACM had an irregularly lumpy shape, and its particle size was about 5–50 μm (Figure 2A). Pure HPBCD was spherical and porous (Figure 2B), and PVPK30 presented a smooth spherical particle (Figure 2C). The physical mixture of ACM/PVPK30/HPBCD (1:18:10) also showed apparent ACM, HPBCD, and PVPK30 particles (Figure 2D). AHP (1:18:10) presented a smooth and layer-like structure, and the obvious irregularly lumpy shape of ACM and the original particle of HPBCD and PVPK30 disappeared (Figure 2E). These findings indicate that ACM nanoparticles had been formed, and ACM was incorporated within HPBCD with PVPK30, resulting in an increase in the surface area of the ACM. Moreover, the TEM image of AHP reconstituted in water shows nano-sized spherical shapes, and its mean particle size was 3.86 ± 0.87 nm (Figure 2F). These results imply that AHP can increase the surface area of raw ACM and improve its water solubility.

### 3.4. Crystalline Transformation of AHP

The results of crystalline transformation using X-ray diffractometry (XRD) analysis are commonly used as one of the strongest forms of evidence to confirm the formation of a drug-delivery system. As shown in Figure 3, there are no particularly characteristic diffraction peaks in the XRD pattern of HPBCD and PVPK30, which indicated that both excipients were in the amorphous state. Our result also shows that raw ACM had obvious characteristic diffraction peaks at 28.3° and 40.3°, indicating that ACM had a crystalline state. The results also showed the same characteristic diffraction peaks in the physical mixture (1:18:10); therefore, PVPK30/HPBCD could not decrease the crystallinity of ACM through the physical mixing process. On the contrary, no diffraction peaks could be seen in AHP (1:18:10), which implied that ACM was successfully incorporated into HPBCD and PVPK30 and formed an inclusion complex nanoparticle due to the amorphous transformation of raw ACM after the nanoparticle process.

### 3.5. Intermolecular Bond Formation

The FTIR results could be used as evidence to elucidate the intermolecular bonds between active ingredients and excipients. A previous study revealed that ACM had many flavonoid compounds, which showed a similar chemical functional group in the FTIR spectrum. As shown in Figure 4, raw ACM displayed several absorptions of functional groups, including at 3382 cm^−1^ (-OH stretching vibration), 2926 cm^−1^ (aliphatic C-H stretching vibration), 1617 cm^−1^ (C=O or C=C stretching), and 1208 cm^−1^ (C-O-C stretching vibration). Obvious absorption of functional groups was observed in the FTIR spectrum of HPBCD, such as at 3389 cm^−1^ (-OH stretching vibration), 2929 cm^−1^ (-CH_2_ stretching vibration), 1643 cm^−1^ (H-O-H stretching vibration), 1153 cm^−1^ (C-O-C stretching vibration), and 1034 cm^−1^ (C-O stretching vibration). The functional groups of OH (3432 cm^−1^), C-H (2956 cm^−1^), C=O (1659 cm^−1^), and C-N (1286 cm^−1^) were present in the FTIR spectrum of PVPK30. In the FTIR spectrum of AHP, the C=O functional group of raw ACM completely disappeared, indicating that the benzopyrone ring with the carboxyl group of the guest had been incorporated into the hydrophobic cavity of HPBCD. The OH group of raw ACM at 3382 cm^−1^ in AHP was shifted towards the higher frequency of 3411 cm^−1^ due to intermolecular hydrogen bonding with HPBCD and PVPK30. These findings imply that raw ACM was successfully incorporated into HPBCD with PVPK30 and formed a nanoparticle formulation. 

### 3.6. In Vitro Skin Penetration of Raw ACM and AHP

The results for the in vitro skin penetration of raw ACM and AHP are displayed in Figure 5. The topical administration of a raw ACM suspension in water was conducted at various time points in pig skin, and the content of ACM in each skin layer was less than 2 μg/cm^2^, indicating that the stratum corneum limited the penetration of the raw ACM suspension to the epidermal and dermal layers. On the contrary, after topical AHP administration, the ACM showed significantly higher penetration through the skin layer than the raw ACM suspension (20-fold) in a time-dependent manner (*p* < 0.05). Figure 5D shows the data in Figure 5C re-expressed as flux-versus-time profiles. It is evident that the flux of raw ACM and AHP gradually declined and then approached a steady-state flux 2 h later for the topical administration of raw ACM and AHP. The skin permeation flux of AHP at 2 h was 71 times higher than that of raw ACM (*p* < 0.05). These findings imply that AHP nanoparticles can overcome the stratum corneum and effectively deliver ACM into deeper skin layers, resulting in increased skin absorption for ACM and increased biological activity.

### 3.7. DPPH Free-Radical-Scavenging Ability of Raw ACM and AHP

The present study used the DPPH free-radical-scavenging assay to validate the antioxidant activity of raw ACM and its nanoparticle formulation. As shown in Figure 6, the highest concentration of raw ACM (500 μg/mL) in water showed about 20% DPPH free-radical-scavenging ability, which is weak. By contrast, AHP significantly scavenged DPPH free radicals in a dose-dependent manner. AHP containing 500 μg/mL of ACM showed 80% DPPH free-radical-scavenging activity, considerably better than that of raw ACM. These results demonstrate that the improvement of the water solubility of ACM could enhance its antioxidant activity in water. Moreover, the manufacturing process for the ACM/PVPK30/HPBCD nanoparticle left the free-radical-scavenging ability of ACM intact.

### 3.8. Cell Viability and Photoprotective Activity of Raw ACM and AHP

It is very important to evaluate the cell safety of an active ingredient and its delivery system before determining its biological activity. As shown in Figure 7A, 1–10 µg/mL of raw ACM in PBS resulted in approximately 100% cell viability in HaCaT cells and exhibited no significant cytotoxicity. However, raw ACM in DMSO and AHP in PBS displayed similar cytotoxicity at 7.5 and 10 µg/mL. The lower concentrations including 1 to 5 µg/mL were used to compare the photoprotective activity. Figure 7B shows the photoprotective activity of raw ACM and AHP. UVB irradiation at a dose of 230 mJ/cm^2^ significantly decreased the cell viability of HaCaT keratinocytes and presented photocytotoxicity (*p* < 0.05). Pretreatment with raw ACM did not present any effect in protecting against HaCaT cell death after UVB overexposure. On the contrary, pretreatment with AHP could significantly prevent the photocytotoxicity in UVB-induced HaCaT cell death (*p* < 0.05). Moreover, overexposure to UVB produced a large amount of ROS and induced oxidative stress, causing HaCaT cell death. Our results show that intracellular ROS production significantly increased in UVB-irradiated HaCaT cells when compared to the unirradiated group (*p* < 0.05). AHP treatment could effectively diminish the overproduction of ROS after UVB irradiation in HaCaT cells, but this effect was not observed for raw ACM (Figure 7C). These results demonstrate that AHP can prevent photodamage through decreasing ROS overproduction.

### 3.9. Cellular Uptake of Raw ACM and AHP

The present study assessed the intracellular uptake of ACM to confirm its photoprotective effect. Pretreatment with raw ACM resulted in trace amounts at 3–24 h in HaCaT cells. Interestingly, pretreatment with AHP resulted in rapid absorption at 3 h, greater absorption at 9 h, and prolonged cellular absorption at 24 h. The cellular uptake at each time point after AHP treatment showed significant cellular absorption compared with that observed for raw ACM treatment (Figure 8). These results reveal that AHP enhanced the penetration of raw ACM into HaCaT cells through decreasing the particle size of the raw ACM.

## 4. Discussion

Natural antioxidants, such as plant extracts [17], vitamins [18], and carotenoids [19], can be active ingredients in skin care products for attenuating oxidative stress and photodamage [20]. *A. communis* methanol extract has been revealed to have great anti-inflammatory and antioxidant activities for preventing UV-induced skin damage [2,21]. The present study indicated that 67% artocarpin was present in ACM, and artocarpin could be an index ingredient for ACM (Figure 1). Tzeng et al. revealed that the water solubility of artocarpin was 0.44 µg/mL [22], making it a practically insoluble drug according to the USP solubility criteria. As shown in Table 1, the water solubility of ACM was 3.50 ± 0.30 µg/mL, indicating that the artocarpin-abundant ACM was a practically insoluble extract. Previous studies have shown that the characteristics of active ingredients with large particle sizes and crystal structures can include lower water solubility, and ACM shows these two characteristics (Figure 2A and Figure 3), which affect its water solubility. In addition, artocarpin has a lipophilic flavonoid skeleton structure, and its physicochemical characteristics are similar to those of widely studied flavonoids, such as quercetin, naringenin, and baicalein, resulting in poor water solubility and permeability [23]. According to the biopharmaceutical classification system, the abundant artocarpin distributed in ACM can be classified as a class II active ingredient. Although most lipophilic active ingredients can be prepared as cream or ointment formulations, the active compounds in ACM did not diffuse thoroughly and showed low distribution in the skin [23]. 

Generally, topical antioxidant nanoparticle formulations can effectively reduce the particle size and change the form from crystalline to amorphous for ingredients showing poor water solubility [24] such as quercetin [25], resveratrol [26], lutein [27], and coenzyme Q10 [28], increasing their dissolution, skin penetration, and bioavailability. Our preliminary test of this study used PVPK30 or HPBCD as an excipient independently to prepare the ACM nanoparticle. Unfortunately, single HPBCD or PVPK30 failed to produce ACM nanoparticles and ACM suspensions, indicating that single HPBCD or PVPK30 cannot include or encapsulate ACM well (data not shown). In the present study, an AHP nanoparticle formulation containing ACM, HPBCD, and PVPK30 was successfully prepared through the solvent-evaporation method. The reason for the formation of the ACM/PVPK30/HPBCD nanoparticle complex is most probably the fact that the drug, CD, and polymer form nanomolecular clusters due to the long-carbon chain PVP solubilization effect for ACM incorporated into HPBCD [29,30]. AHP was also uniformly dispersed as nanomolecular clusters in aqueous solution under TEM observations (Figure 2F). Additionally, our results demonstrate that AHP could obviously increase the water solubility by about 289 times when compared to raw ACM (Table 1). Moreover, the results from the SEM (Figure 2E) and TEM (Figure 2F) indicate that AHP displayed a nanoparticle form and had a greater surface area than raw ACM (Figure 2A). In addition, Baghel et al. mentioned that amorphous solid dispersion could transform the crystalline form of the active ingredient into an amorphous state with a disordered structure, exhibiting a higher free energy, resulting in higher apparent water solubility, dissolution, and oral absorption [31]. The XRD pattern of AHP showed an obvious crystalline form of ACM that was transformed to amorphous and supported the increased water solubility of raw ACM (Figure 3). Intermolecular bonds such as H-bonding, van der Waals forces, electrostatic bonds, ionic bonds, or hydrophobic bonds restricted the molecular mobility of the active ingredients in the carriers and provided stability to the system. Khougaz et al. revealed that the intermolecular forces between the PVP carbonyl group and the MK-0591 carboxylate group can play an important role in stabilizing amorphous drugs in solid dispersions of MK-0591 with PVPK12 [32]. In our study, the FTIR spectra also revealed that ACM was incorporated into HPBCD with PVPK30, and a certain degree of interaction between ACM and carriers, for the stability of amorphous ACM, is important for forming a novel nanoparticle formulation (Figure 4). Similar findings were also found for kelp phlorotannin [33], indomethacin [34], and fullerenes [35]. These changes in physicochemical characteristics could elucidate the mechanism of water-solubility improvement. 

It is well known that the stratum corneum is the most important skin barrier layer, affecting the skin absorption of many active ingredients and reducing their biological activities. Topical nanoparticle antioxidant formulations are one of the cutaneous delivery systems for enhancing the cutaneous absorption of antioxidants for preventing many skin problems [36]. Our results show that raw ACM could not only not penetrate the stratum corneum, but also was not deposited in the epidermis and dermis, indicating that the stratum corneum prevents ACM from being absorbed by the skin due to the poor water solubility of raw ACM (Figure 5). On the contrary, the significant enhancement of ACM permeated in AHP contributed to the presence of HPBCD with PVP (Figure 5), indicating that ACM can be delivered via an HPBCD with PVPK30 nanocarrier through the stratum corneum, with the possibility of exerting local effects to prevent skin diseases. There are three possible reasons for the enhancement of skin penetration after AHP topical administration. First, the skin-permeation enhancement was correlated with an improvement in the water solubility of the raw ACM. HPBCD and PVPK30 not only represent a common solubilizer enhancing the solubilization of ACM but also stabilized ACM in AHP or at the skin layer, resulting in the enhancement of the skin permeation of ACM. A similar finding was also found for ethyl 4-biphenyl acetate [37] and lornoxicam [38] using HPBCD as a skin enhancer for improving skin permeation. Second, the diameter of the AHP nanoparticles was 3.86 ± 0.87 nm, allowing them to easily penetrate the stratum corneum and to deposit in sweat ducts and sebaceous glands; therefore, AHP can increase the skin absorption of ACM by intercellular and appendage penetration pathways. Third, Pünnel and Lunter revealed that HPBCD and PVP are polymers commonly used in film-forming systems [39] and can form a thin-film layer on the skin surface. This phenomenon is called the skin occlusive effect, which can effectively help active ingredients to penetrate the stratum corneum and increase skin absorption through skin hydration, enabling their biological activity. Taken together, AHP, as a good non-invasive method, can significantly increase the skin penetration of ACM by overcoming the skin barrier. 

Nanoparticle formulation systems can effectively reduce the particle sizes of active ingredients and change crystalline forms to amorphous; however, this results in extra Gibbs free energy and lower stability than microparticle formulation systems due to the recrystallization of active ingredients and changes in surface energy [31]. Additionally, environmental factors in the pharmaceutical manufacturing process, such as light and temperature, might affect the pharmacological effects of antioxidants, limiting their applications in medicine and cosmetics [40]. An extract of *A. communis* in an organic solvent displayed a DPPH scavenging effect and could effectively protect the skin against UVB in vitro and in vivo through antioxidant, anti-inflammatory, and anti-apoptotic effects [41,42]. Therefore, we performed a DPPH assay and photoprotection assay in keratinocytes to confirm the biological activity of AHP and raw ACM. The present study demonstrated that AHP containing 500 μg/mL of ACM showed DPPH free-radical-scavenging activity that was four times higher than that of raw ACM (Figure 6). AHP exerted photoprotective activity through reducing the ROS content in UVB-induced keratinocyte damage (Figure 7). On the other hand, our results further demonstrate that AHP can significantly increase the cellular uptake of ACM compared to raw ACM, resulting in better photoprotective effects. By contrast, raw ACM showed no photoprotective or antioxidant activity, due to the lower cell absorption of ACM. These findings also reveal that the AHP nanoparticle can easily diffuse in the intercellular space and then become attached to the cell membrane, resulting in increased cell absorption (Figure 8). Brough and Williams mentioned that solid amorphous dispersions (SADs) are a common pharmaceutical method used to increase the apparent solubility of crystalline drugs, and SADs can also prevent the recrystallization of active ingredients, stabilizing amorphous drugs in a polymer matrix through the formation of weak intermolecular forces between the drug and polymer during the storage of an amorphous formulation [43]. HPBCD is a good stabilizer with which to stabilize the amorphous states of active ingredients, including alpha-lipoic acid [44] and dihydroartemisinin [45], and PVPK30 can also provide superior stabilization for lacidipine [46] and loperamide [47]. Our data support the idea that HPBCD with PVPK30 not only enhances solubility but also stabilizes the amorphous form of AHP, maintaining the biological activity of ACM due to ACM/HPBCD/PVPK30 polymer interactions in solution.

## 5. Conclusions

The present results demonstrate that *A. communis* methanol extract was successfully incorporated into PVPK30 with HPBCD to form nanoparticles, significantly improving the water solubility of ACM through reducing the particle size and inducing amorphous transformation. AHP can also effectively increase skin absorption and cellular uptake due to ACM particle nanosization, resulting in enhanced photoprotective and antioxidant activity. Consequently, AHP nanoparticles could be used as an antioxidant additive for preventing UV-induced skin problems.

## Figures and Tables

**Figure 1 pharmaceutics-13-01372-f001:**
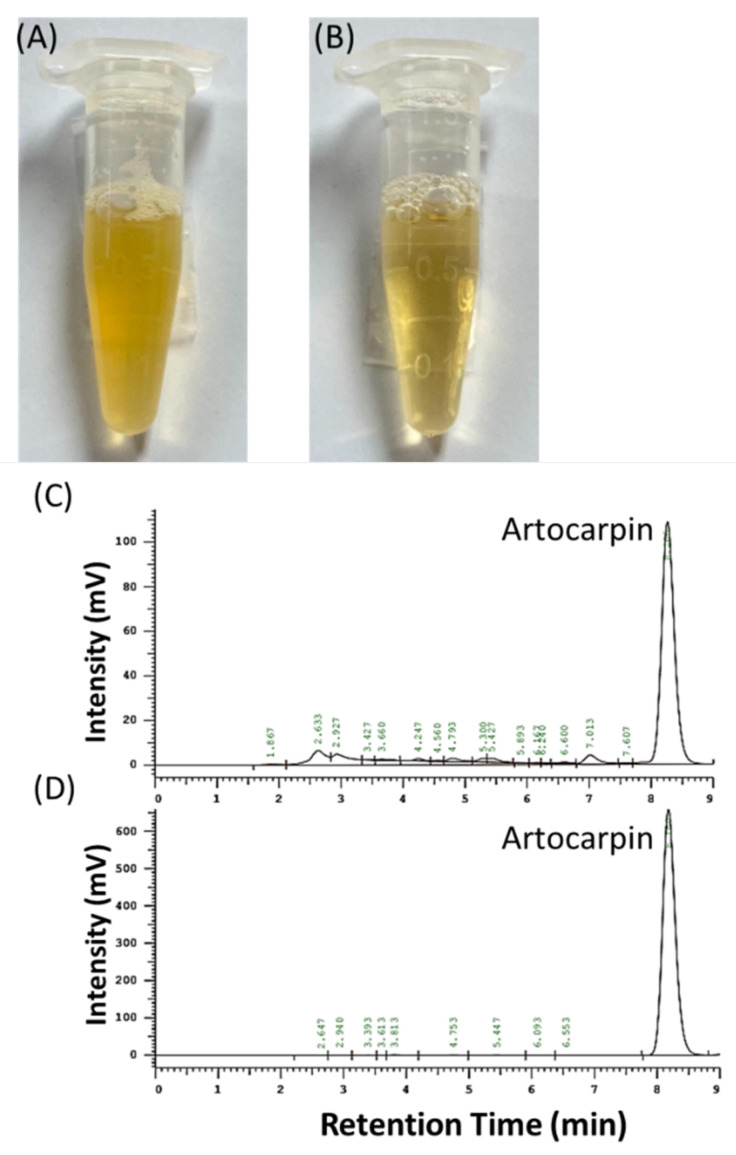
Photographs of (**A**) raw ACM and (**B**) AHP (1:18:10) dispersed in pure water, and high-performance liquid chromatography (HPLC) profile of (**C**) *Artocarpus communis* methanol extract (ACM) at 100 µg/mL and (**D**) artocarpin at 200 µg/mL (index component).

**Figure 2 pharmaceutics-13-01372-f002:**
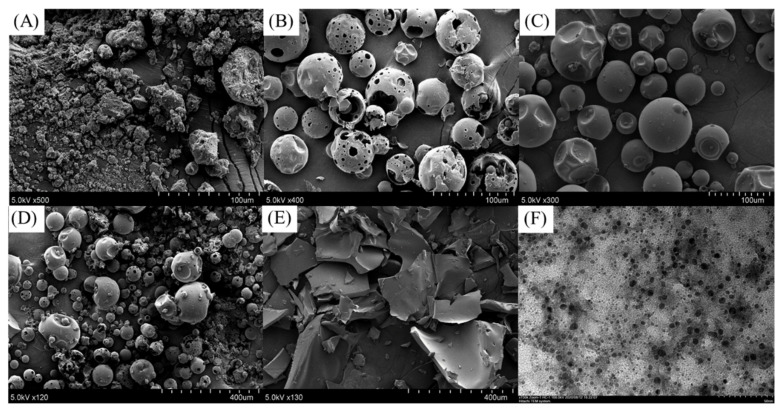
SEM images of (**A**) raw ACM, (**B**) HPBCD, (**C**) PVPK30, (**D**) physical mixture ACM/PVPK30/HPBCD (1:18:10), and (**E**) AHP 1:18:10 and (**F**) particle morphology of AHP 1:18:10 under TEM observation.

**Figure 3 pharmaceutics-13-01372-f003:**
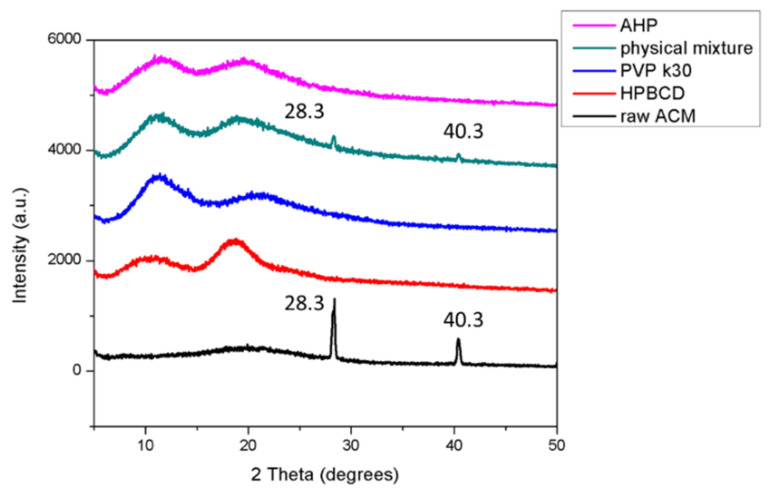
Powder X-ray diffraction patterns of ACM, HPBCD, PVPK30, physical mixture, and AHP.

**Figure 4 pharmaceutics-13-01372-f004:**
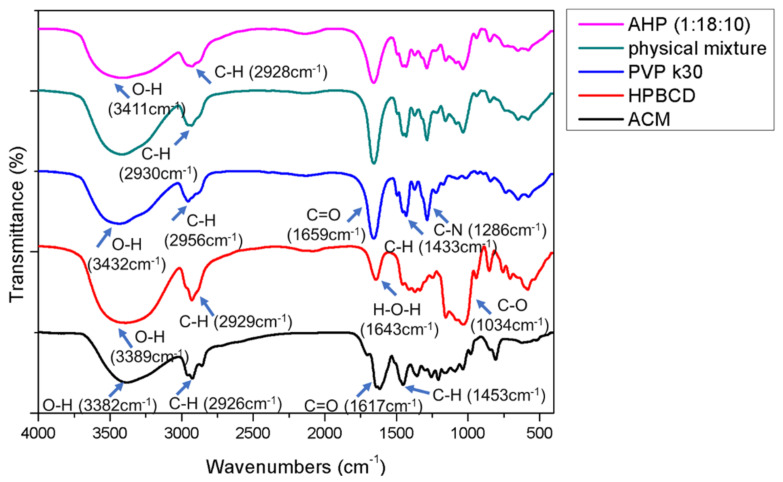
Fourier transform infrared (FTIR) spectra of raw ACM, HPBCD, PVPK30, physical mixture, and AHP.

**Figure 5 pharmaceutics-13-01372-f005:**
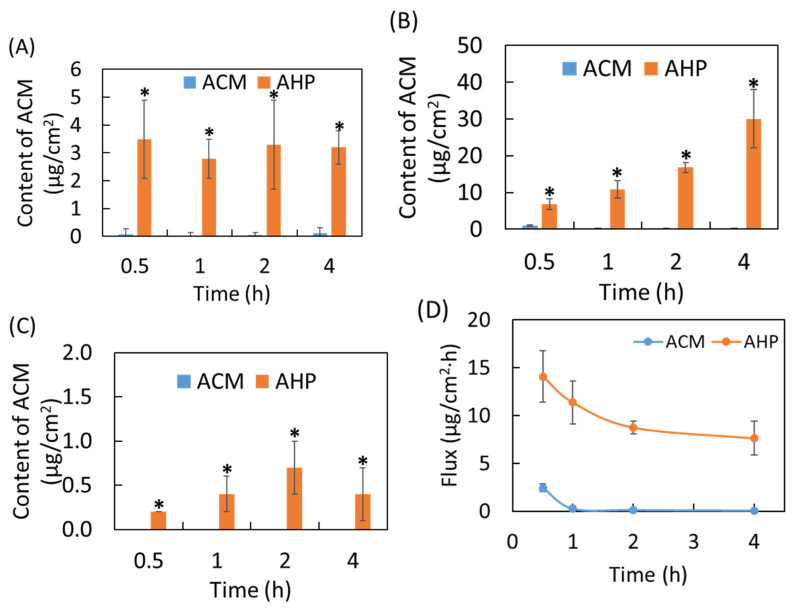
In vitro skin penetration of raw ACM and AHP: (**A**) stratum corneum, (**B**) epidermis, (**C**) dermis, and (**D**) flux of raw ACM and AHP. Results are expressed as mean ± SD (*n* = 5). * represents *p* < 0.05 when compared to raw ACM.

**Figure 6 pharmaceutics-13-01372-f006:**
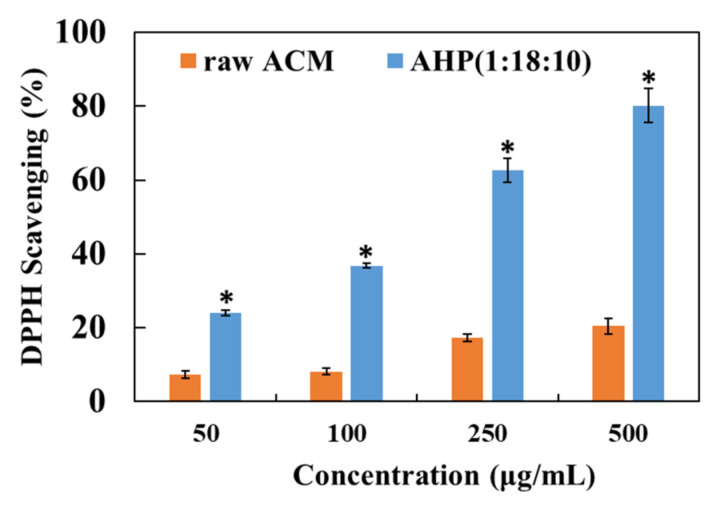
The DPPH free-radical-scavenging activity of raw ACM and AHP nanoparticle reconstituted in deionized water. * represents *p* < 0.05 when compared to raw ACM.

**Figure 7 pharmaceutics-13-01372-f007:**
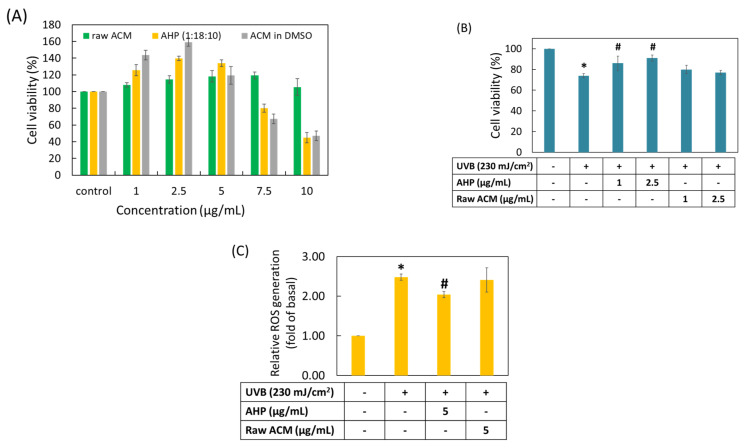
Cell viability and photoprotective activity of raw ACM and AHP in HaCaT keratinocytes. (**A**) cell viability of HaCaT cells treated with raw ACM in PBS or DMSO, or AHP in PBS; (**B**) photoprotective activity of raw ACM and AHP in UVB-induced HaCaT keratinocyte damage; (**C**) ROS production under UVB irradiation of HaCaT cells treated with raw ACM and AHP. Data are from three independent experiments. Values are expressed as mean ± SD (*n* = 3). * *p* < 0.05 compared with the control group, and # *p* < 0.05 compared with the UVB-exposure group. “+” Indicated cells treated with UVB irradiation, “-“ Indicated cells without treatment.

**Figure 8 pharmaceutics-13-01372-f008:**
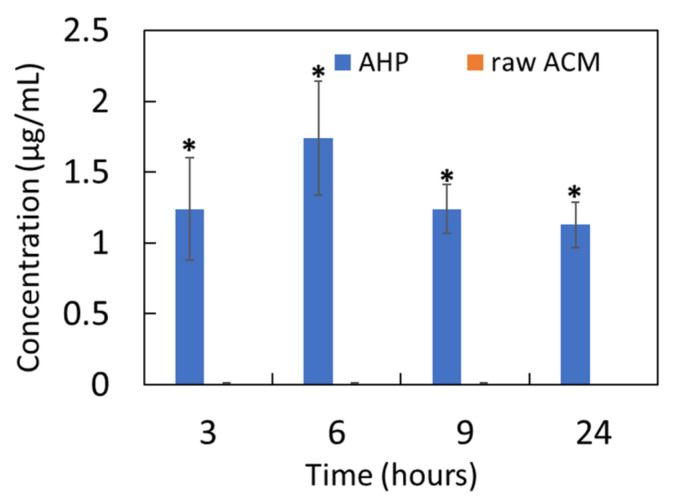
Cellular uptake of raw ACM and AHP at 5 μg/mL. * represents *p* < 0.05 when compared to raw ACM.

**Table 1 pharmaceutics-13-01372-t001:** The formulation and water solubility, yield, and encapsulation efficiency of AHP.

ACM:PVP:HPBCD	Water Solubility (µg/mL)	Yield (%)	Encapsulation Efficiency (%)
1:18:2	484.61 ± 1.05 *^,#^	85.43 ± 4.15	91.8 ± 6.1
1:18:10	1012.42 ± 37.4 *	92.15 ± 4.06	99.9 ± 0.1
1:18:20	659.16 ± 1.42 *^,#^	83.56 ± 0.78 ^#^	88.2 ± 2.3 ^#^
Raw ACM	3.50 ± 0.30	-	-

Data are expressed as the mean ± standard deviation (SD), and all the data were collected in triplicate. * *p* < 0.05 compared with raw ACM, and ^#^
*p* < 0.05 compared with AHP 1:18:10.

## Data Availability

All data presented in the study are available on request from the corresponding author (flyen@kmu.edu.tw).

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
