# Peer review of "Topical Artocarpus communis Nanoparticles Improved the Water Solubility and Skin Permeation of Raw A. communis Extract, Improving Its Photoprotective Effect"

_pharmaceutics, 2021, doi:10.3390/pharmaceutics13091372_

Round 1

Reviewer 1 Report

The manuscript deals with the preparation of particles based on hydroxypropyl-β-cyclodextrin and polyvinylpyrrolidone K30 loaded with Artocarpus communis methanol extract (ACM). The final aim is the improvement of water solubility and skin penetration; some biological activities were also verified.

Overall, along the manuscript it is unclear why discussion is relies on ACM instead of the main component, the active ingredient Artocarpin. The technological characterization of the particles should be discussed in this way.

The text requires extensive revision of the English form.

I have some questions and comments on the experimental work, the discussion and the format, as well. Therefore, I recommend its publication after major revision.  

Introduction.

Line 45-48. It should be better explained what the authors mean. Particles containing an extract, as such, are not a formulation for topical application.

Line 60-61: add data or reference.

“Materials and methods”

  • 1 Materials. Artocarpin is nowhere mentioned.
  • 3 HPLC analysis of ACM. Explain why series concentration of ACM and artocarpin were prepared.
  • 4 Preparation of ACM loaded PVPK30/HPBCD nanoparticle (AHP). Used amounts of HPBCD and PVPK30 should be explained in more detail.
  • 6 Yield and encapsulation efficiency. Method for determining encapsulation efficiency should be explained in more detail.
  • 7 Morphology and particle size analysis. Check line 141 (water dilution).
  • 10 DPPH free radical scavenging activity. Check line 164 (OD value).
  • 11 In vitro skin penetration. Tape stripping should be explained in more detail.
  • 12 Cell viability. Check typos and add description of abbreviations.

“Results”

  • 1 The yield, artocarpin content, and calibration curve of ACM. Line 229: no previous mention to lyophilization is reported. Line 236: in my opinion it is confusing to discuss the results in term of ACM. I recommend switching to artocarpin content.
  • 2 Water solubility, yield, encapsulated efficiency of AHP. Line 245-246: already written. Line 246: it is not described how the mentioned ratios were selected.
  • 3 Morphology of ACM and AHP. Description and results of Figure 2E and 2F should be explained in more detail.
  • 12 Cell viability and photoprotective activity of raw ACM and AHP. Check legend Figure 7.

“Discussion”

Too simple and sometimes repetitive; it should be combined with the results.

Author Response

Dear review:

Reviewer 2 Report

Comments:

The manuscript addresses an interesting research topic; the results were well presented and discussed. The English must be revised. The discussion could be improved, a lot of recent research have been made on skin permeability and UV protection. The last topic was poorly discussed. At present, there are some lacking points, which will be important to clarify:

Line 88: “…dried in the shade..” specify the temperature and time. Would be interesting to have an idea about the sample moisture. In the same line “…200 g of plant pieces…”identify better which part of the plant was used.

There any extraction optimization procedure that allowed to achieve the conditions of 200g sample for 4L solvent?

The HPLC analysis was only made for artocarpin quantification? No other standards were injected for a complete characterization of the sample?

Is not clear how author’s made a calibration curve of the ACM extract. Did author’s refer to the artocarpin content in the extract?

Line 173 indicate the skin diameter that was exposed to the sample.

Line 174 if the concentration of AHP was 1mg/mL the extract amount tested in this case was inferior to the raw extract. Clarify what is (1:18:10) and the units?

If the solubility of artocarpin in water in 3.5 ug/mL when author’s applied 1mg/mL in skin what was the effective applied dose?

Table 1. Statistical analysis that show significant differences between formulations is missing.

Figure 3: typing error “aw”.

Line 306: “in vitro” in italic

Line 311: “p< 0.05” p should come in italic and space between numbers and units.

Figure 5.D. The cumulative amount is normally referred to the permeated amount into the receptor chamber and not in the skin as described. Statistical analysis.

On the skin permeation author’s did not mention if the recovered amount was equally to the applied dose.

Figure 6. Statistical analysis.

Figure 7.A should be just Figure 7.

If AHP in PBS displayed cytotoxicity at 7.5 μg/mL and 10 μg/mL what was the purpose of testing permeability with 1m/mL. How author’s can guarantee that concentration was not disturbing the skin layers structure affecting the permeability kinetic?

Author Response

Dear reviewer:

Round 2

Reviewer 1 Report

Accept in the present form.

Reviewer 2 Report

The paper os acceptable for publication.